# Real-Time In Vitro Fluorescence Anisotropy of the Cyanobacterial Circadian Clock

**DOI:** 10.3390/mps2020042

**Published:** 2019-05-24

**Authors:** Joel Heisler, Archana Chavan, Yong-Gang Chang, Andy LiWang

**Affiliations:** 1Chemistry & Chemical Biology, University of California, Merced, CA 95343, USA; jheisler@ucmerced.edu; 2Center for Cellular and Biomolecular Machines, University of California, Merced, CA 95343, USA; 3School of Natural Sciences, University of California, Merced, CA 95343, USA; achavan@ucmerced.edu (A.C.); Tyler.Chang@monash.edu (Y.-G.C.); 4Center for Circadian Biology, University of California, San Diego, La Jolla, CA 92093, USA; 5Quantitative & Systems Biology, University of California, Merced, CA 95343, USA; 6Health Sciences Research Institute, University of California, Merced, CA 95343, USA

**Keywords:** fluorescence, circadian clock, cyanobacteria, protein, phosphorylation

## Abstract

Uniquely, the circadian clock of cyanobacteria can be reconstructed outside the complex milieu of live cells, greatly simplifying the investigation of a functioning biological chronometer. The core oscillator component is composed of only three proteins, KaiA, KaiB, and KaiC, and together with ATP they undergo waves of assembly and disassembly that drive phosphorylation rhythms in KaiC. Typically, the time points of these reactions are analyzed ex post facto by denaturing polyacrylamide gel electrophoresis, because this technique resolves the different states of phosphorylation of KaiC. Here, we describe a more sensitive method that allows real-time monitoring of the clock reaction. By labeling one of the clock proteins with a fluorophore, in this case KaiB, the in vitro clock reaction can be monitored by fluorescence anisotropy on the minutes time scale for weeks.

## 1. Introduction

Organisms from all domains of life display circadian (~24 h) rhythms in their metabolism, physiology, and behavior that arose as an adaptation to daily cycles of ambient light and temperature [1]. These endogenous rhythms are generated by intracellular circadian clocks. Despite extensive investigations in fungi, plants, insects, and vertebrates, the mechanistic nature of circadian clock protein-protein interactions remains mysterious. Among model systems, the cyanobacterial clock offers a unique opportunity in this regard. It can be reconstituted in vitro by simply mixing its three protein components—KaiA, KaiB, and KaiC—with ATP, resulting in a macroscopic ~24 h rhythm of KaiC phosphorylation [2]. 

Just as a watchmaker’s apprentice learns the mechanism of a watch by studying its gears as they move, it is informative to observe components of the cyanobacterial clock as they move. Typically, time points of in vitro cyanobacterial clock reactions are analyzed ex post facto using denaturing polyacrylamide gel electrophoresis (SDS PAGE) to resolve different states of KaiC phosphorylation. It was used to resolve the ordered temporal pattern of KaiC phosphorylation: S/T→S/pT→pS/pT→pS/T→S/T→…, where S and T represent residues S431 and T432, the two phosphorylation sites of KaiC, and pS and pT denote their phosphorylated states [3,4]. This method has allowed for numerous insights into the cyanobacterial clock, such as KaiA stimulates KaiC autophosphorylation during the day, and KaiB inhibits KaiA in order to promote KaiC autodephosphorylation at night [5,6,7,8,9,10]. However, it has some inherent disadvantages. For example, reactions cannot be monitored in real time. Removing aliquots from reactions every few hours followed by SDS PAGE limits how many experiments can be run in parallel and the duration of each experiment. Electrophoresis and densitometry of stained gels to determine KaiC phosphorylation levels is a manual process. In addition, this SDS PAGE approach has low temporal resolution (hour time scale), ≥10% uncertainty in KaiC phosphorylation levels per time point, does not directly inform on protein-protein interactions, and perturbs the sample (by taking time points). However, the recent development of an automated sampling device could reduce some of these drawbacks [11].

Here, we demonstrate that in vitro fluorescence spectroscopy can directly monitor circadian rhythms of protein-protein interactions in the cyanobacterial clock in real time by utilizing clock proteins labeled with fluorescent dyes. This methodology uses a fluorescently-labeled construct of KaiB [12,13,14], and the plasmid construct for expressing KaiB-FLAG-K25C (described in the Section 2.1.1 and Section 3.1) is available to the scientific community. It is worth mentioning here that real-time measurement of luciferase-based bioluminescence in vivo is a powerful method for investigating circadian gene expression rhythms in cyanobacteria [15]. The availability of crystal structures of the free clock components and their complexes make it feasible to select fluorophore labeling sites that do not perturb the system [16]. As will be demonstrated below, tracking fluorescence anisotropy of 6-iodoacetamidofluroescein (6IAF) labeled KaiB allows direct observations of real-time population shifts between free KaiB (daytime) and bound KaiB (nighttime) (Figure 1). This fluorescence method is straightforward, does not perturb the sample during measurements, and offers a high time resolution (minutes) of the clock as it ticks.

The methodology utilizes standard molecular cloning, protein expression, and labeling procedures. A convenient approach to site-directed DNA mutagenesis is Quikchange PCR, allowing for substitutions, deletions, or small additions with a single-step polymerase chain reaction conducted on a plasmid vector [18]. Then, standard heat shock transformation of *E. coli*, expression, purification of the protein and fluorophore labeling sets the stage for facile real-time measurements of protein-protein interactions in oscillating clock reactions. Figure 2 provides structural insight into the selected KaiB conjugation site for fluorophore labeling, highlighting how lysine 25 is oriented away from the intramolecular tetramer and dimer interfaces, and intermolecular KaiC and KaiA binding sites, allowing for conserved function of KaiB. 

Fluorescence anisotropy is widely used by biochemists. Basically, shining polarized light on an isotropic solution selectively excites fluorophores whose transition moments are parallel to the direction of polarization [19]. Excited-state lifetimes of many commercially available fluorophores are similar to the rotational correlation times of proteins (nanoseconds), making fluorescence anisotropy of labeled proteins sensitive to protein-protein interactions (Figure 1). 

## 2. Experimental Design

### 2.1. Materials

#### 2.1.1. Quikchange Preparation of the KaiB-K25C Construct for Fluorescence Labeling

KaiB-K25C plasmid construct [14] (available upon request)10 ng/µL pET28b+ plasmid template DNA of *kaiB* gene.Critical step: We used the pET28b+ bacterial expression vector, which produces N-His_6_-SUMO-proteins. Kanamycin resistance is important for several steps outlined in this protocol. If other vectors are used, adjustments to the antibiotic steps throughout this protocol may be needed.DpnI restriction enzyme (10 units/µL) (New England BioLabs; Cat. no.: R0176L, Ipswich, MA, USA).Quikchange PCR primers.Critical step: Quikchange PCR primers for *Synechococcus elongatus* KaiB-FLAG-K25C:Forward-CCAAACTCAGTCCGTGCCCTCTGCACGCTCAAGAACATTCTCGReverse-CGAGAATGTTCTTGAGCGTGCAGAGGGCACGGACTGAGTTTGG10× PfuTurbo Cx reaction buffer (1.0 units/50 µL PCR) (Agilent Technology; Cat. no.: 600257).PfuTurbo DNA polymerase (1.0 units/50 µL PCR).10 mM deoxynucleotide triphosphates (dNTPs) solution mix (New England BioLabs; Cat. no.: N0447L, Ipswich, MA, USA).1.5% agarose gel in Tris base (Fisher Chemical; Cat. no.: BP152-10, Waltham, MA, USA), acetic acid (Fisher Chemical; Cat. no.: A38-k212, Waltham, MA, USA), and EDTA (Fisher Scientific; Cat. no.: BP121-500), referred to as (TAE) buffer.Competent BL21-DE3 *E.coli* cells.Luria Broth (LB) agar plates with 50 µg/mL kanamycin (ThermoFisher Scientific, Cat. no.: 11815032, Waltham, MA, USA, USA).Critical step: Autoclave 1 L of Luria broth (LB) media broth in 2 L Erlenmeyer flask on liquid cycle. Allow to cool to 50 °C and add 1 mL (50mg/mL) kanamycin to the sterilized 2 L LB flask. Pour into sterile plastic plates and immediately cover with lids. Let sit at room temperature for 20 min before wrapping in aluminum foil and storing at 4 °C until use. We find that these plates are good for up to one month.500 µL of 2× YT bacterial culture media broth per sample.

#### 2.1.2. PCR Screening and DNA Sequencing

LB kan^+^ media broth.1.5% agarose gel in Tris base, acetic acid, and EDTA (TAE) buffer.

#### 2.1.3. Protein Expression

Freshly transformed cells harboring the expression construct.LB media to grow cells.50 mg/mL kanamycin antibiotic.1 M isopropyl β-D-1-thiogalactopyranoside (IPTG) (Research Products International; Cat. no.: I56000-200.0, Mt Prospect, IL, USA).Lysis buffer (50mM NaH_2_PO_4_ (Sigma-Aldrich; Cat. no.: S3139-500G, St. Louis, MO, USA), 500mM NaCl (Fisher Chemical; Cat. no.: S271-10, Waltham, MA, USA), pH 8.0).Critical step: Lysis buffer, wash buffer, and elution buffers are best stored at 4 °C until use. Elution buffer should avoid light, because the high concentration of imidazole turns the solution yellow with light exposure. However, slight discoloration does not affect the elution of His-tag proteins.For proteins to be fluorescently labeled, tris(2-carboxyethyl)phosphine (TCEP) (GoldBio; Cat. no.: TCEP25, St Louis, MO, USA) was used to prevent unwanted disulfide bond formation between proteins.Critical step: Proteins can crosslink through cysteinyl –SH groups. To minimize this unwanted reaction, a final concentration 20 mM TCEP was added to the elution volume following Ni-NTA column purification. More TCEP can be added after the 2nd Ni-NTA column purification step, preceding concentration of the sample for FPLC (see steps 55–62). No additional TCEP is added following size-exclusion chromatography as to not interfere with 6IAF labeling efficiency.

#### 2.1.4. Fluorescent Labeling of Clock Protein

Fluorescent dye compatible with thiol (-SH) click chemistry attachment (6-iodoacetamidofluorescein) (ThermoFisher Scientific; Cat. no.: I30452, Waltham, MA, USA).Critical step: Here, for each fluorescently labeled protein preparation, 1 mg of 6IAF was suspended with 80 µL of methanol and then mixed in a 5:1 ratio with KaiB-FLAG-K25C at pH 7.0 to facilitate conjugation (see step 72).Reduced KaiB-K25C mutant construct with cysteinyl residue available for fluorophore labeling (wild-type KaiB has no naturally occurring cysteinyl residues).Labeling buffer (20 mM Tris, 150mM NaCl, pH 7.0).Critical step: After labeling, a desalting column is used to both adjust pH to 8.0 for oscillation reactions while also removing free dye from the sample.Desalting column buffer for protein storage (20 mM Tris, 150 mM NaCl, pH 8.0).

#### 2.1.5. Fluorescence Anisotropy Binding Assay

Fluorescently labeled clock protein (see Section 3.5).600 µL reaction buffer (20 mM Tris, 150 mM NaCl, 5 mM MgCl_2_ (Fisher Scientific; Cat. no.: BP214-500, Waltham, MA, USA), 1 mM ATP (MP Biomedicals; Cat. no.: 194613O, Burlingame, CA), 0.5 mM EDTA, 50 µg/mL kanamycin, pH 8.0) for each reaction sample.Unlabeled cyanobacterial clock proteins FLAG-KaiC, KaiA, KaiB-FLAG [20] (plasmids available upon request).Critical step: Here, we fused N- and C-terminal FLAG-tags (DYKDDDDK) to KaiC and KaiB proteins, respectively, to increase solubility and stability.Low protein-binding syringe filter (13 mm with 0.2 um membrane).

### 2.2. Equipment

#### 2.2.1. Quikchange Preparation of the KaiB-K25C Construct for Fluorescence Labeling

Micropipettes (0.2–2 µL, 2–20 µL, and 20–200 µL) and pipette tips.DNA molecular size ladder (New England BioLabs; Cat. no.: N3200L, Ipswich, MA, USA).0.2 mL individual PCR tubes with snap shut caps to hold PCR reactions.Thermocycler.Gel electrophoresis apparatus (BIO-RAD Wide Mini-Sub Cell GT and PowerPac universal power supply, Hercules, CA, USA).1.5 mL sterile centrifuge tubes.Ice bath (small bucket or tray).Water bath (set to 42 °C).Temperature controlled shaker (set at 37 °C).Glass rod for spreading BL21-DE3 transformed cells on LB agar plates.Alcohol lamp or Bunsen burner for sterile technique.Incubator (set at 37 °C) for cultured LB agar plates.

#### 2.2.2. PCR Screening and DNA Sequencing

Micropipettes (0.2–2 µL, 2–20 µL, and 20–200 µL) and pipette tips.15 mL sterile Falcon tubes.DNA molecular size ladder.Thermo Scientific GeneJET Plasmid Miniprep Kit (ThermoFisher Scientific; Cat. no.: K0503, Waltham, MA, USA).Benchtop centrifuge.Gel electrophoresis apparatus (BIO-RAD Wide Mini-Sub Cell GT and PowerPac universal power supply, Hercules, CA, USA).Thermo Scientific Sorvall Legend RT+ Centrifuge.Thermo Scientific NanoDrop UV-Vis 2000 Spectrophotometer.Temperature-controlled shaker at 37 °C and capable of 220 rpm.

#### 2.2.3. Protein Expression

Inoculating loop.Autoclave.UV-Vis spectrometer (HACH DR 6000EDU, Loveland, CO, USA).Temperature controlled shaker (set at 37 °C).Centrifuge to harvest cells, and to separate protein supernatant from insoluble debris after cracking cells. We use a Thermo Scientific Sorvall RC 6 Plus Superspeed centrifuge. It has 4 L of capacity, speeds of up to 22,000 rpm, and RCF’s to 55,200× *g*.Centrifuge rotors for 50 mL and 300 mL centrifuge bottles.Sterile volumetric pipet.Transfer bulb for volumetric pipet.Homogenizer for cell disruption.Critical step: An Avestin C3 homogenizer was used to produce homogenous cell-disrupted samples. Allowing the samples to cycle through the instrument for ~5 min was sufficient to consistently achieve homogenous products. Subsequent centrifugation yielded clear separation of supernatant containing soluble proteins and cell debris pellet.Nickel-nitrilotriacetic acid (Ni-NTA) gravity-flow chromatography columns for protein purification (10 mL polypropylene columns with QIAGEN Ni-NTA agarose beads) (Qiagen; Cat. no.: 30230).Amicon stirred cell and a 10 kDa cut-off ultrafiltration disc (Millipore Sigma; Cat. no.: PLGC04310, St. Louis, MO, USA).Benchtop centrifuge and 1.5 mL centrifuge tubes.Liquid chromatography 5 mL injection syringe, needle, and adaptor for injection into instrument.Fast protein liquid chromatography (FPLC) to collect protein eluates.GE Healthcare Life Sciences size-exclusion HiLoad 16/600 Superdex 75.Nanodrop to measure protein concentrations by Bradford Assay.Critical step: Protein concentrations were determined with bovine serum albumin (BSA) standard curve Bradford Assay. A standard curve was made fresh each time a protein was measured.

#### 2.2.4. Fluorophore Labeling of Clock Protein

Covered sterile Corning tube to hold protein sample during labeling reaction without light exposure.Critical step: Simply wrapping the corning tubes with aluminum foil, fixed with tape or parafilm is sufficient to protect the fluorescent dye from light exposure during labeling.FPLC to separate labeled KaiB construct from free dyes.Stirred cell concentrator.Critical step: The concentrator employed here was Amicon stirred cell 50 mL concentrator with a 10 kDa molecular weight cutoff regenerated cellulose membrane. This apparatus was connected to a nitrogen gas line to provide pressure, and magnetic stir plate allowed for stirring while concentrating. Concentration occurred in a 4 °C refrigerator.Liquid chromatography 2 mL injection syringe, needle, and adaptor for injection into instrument.GE Healthcare Life Sciences HiPrep 26/10 desalting column.Nanodrop to measure protein concentrations by Bradford Assay.

#### 2.2.5. Fluorescence Anisotropy Binding Assay

Photon counting spectrofluorimeter.Critical step: ISS Photon Counting Spectrofluorimeter (PC1) shown in Figure 3, with 300 W xenon arc lamp attenuated such that fluorescently-labeled samples yield 600,000–800,000 photon counts, was used for all fluorescence measurements described here. This steady-state fluorimeter was setup with a three-stage cuvette system and water bath to control sample temperature. Excitation and emission wavelengths are easily manipulated, allowing for measurements of multiple fluorescent dyes simultaneously.Three-cuvette sample compartment for fluorimeter.Circulating water bath at 30 °C and tubing compatible with fluorimeter for sample temperature control.5 mm quartz cuvettes to hold reaction samples.Low protein-binding 0.2 µm membrane filter for each reaction sample (Pall Corporation; Cat. no.: PN 4602).1 mL syringe for sample filtration.Stock MgCl_2_ and ATP solutions.Critical step: ATP and MgCl_2_ stocks at concentrations of 300 mM and 200 mM, respectively, were used. These should be prepared in 20 mM Tris with pH 8.0. Note that KaiA and KaiB protein stocks lack ATP and MgCl_2_.1.5 mL sterile centrifuge tubes, three for each reaction sample (see steps 80–88).

## 3. Procedure

### 3.1. DNA Sample Preparation using Quikchange PCR (Time of Completion: 5–6 h)

1.Prepare Quikchange PCR solutions in PCR tubes (see Table 1).2.Setup thermo cycler to run program (see Figure 4).Critical step: The PCR products should be run on 1.5% agarose gels before performing DpnI digestion. Load 3 µL PCR product and 3 µL 6× orange loading dye. Mix well with micropipette and heat to 100 °C for 1 min before loading agarose gel. Only add DpnI to the products that show bright bands at the correct size of the plasmid.3.Add 1 µL of DpnI enzyme to each Quikchange PCR product, and spin down in benchtop centrifuge at 13,000 rpm for 1 min.4.Incubate Quikchange PCR products with DpnI for 1 h at 37 °C.5.PCR purification kit, optional (QUIAGEN).6.Measure Quikchange PCR product concentration with Nanodrop. Ensure the 260 nm/280 nm ratio is near 1.8 for pure DNA product.Critical step: Sample purity can be obtained by comparing the absorbance at 260 nm against that of 280 nm (A_260/280_). Highly pure DNA in should be ~1.8, while highly pure RNA should be ~2.0. Proteins absorb light at 280 nm. Therefore, lower ratios are indicative of protein contamination in the sample [22].

### 3.2. Transformation of Competent Cells with DpnI-Treated Quikchange Products (Time of Completion: 1 Day)

7.Chill centrifuge tubes on ice for 5–10 min.8.Warm 500 µL of 2× YT broth for each transformation sample.9.Add 50 µL of BL21-DE3 competent cells to each chilled centrifuge tube.10.Add 2 µL of DpnI digested Quikchange PCR product, mix well and incubate on ice for 30 min.Critical step: For the final five minutes of the 30 min period in which the Quikchange PCR product is on ice, place the 2× YT media broth stock tubes in the hot water bath at 42 °C. Remove the 2× YT media broth from the hot water bath at the same time the heat shock ends (see step 11).11.Heat shock the cells at 42 °C in a water bath for 45 s.12.Incubate them on ice for 2 min.13.Add 500 µL of warmed 2× YT broth to each sample.14.Incubate cells in temperature-controlled shaker at 37 °C and 220 rpm for 1 h.15.Spin down cells at 5000 rpm for 1 min in benchtop centrifuge (at room temperature).16.Discard 350 µL of supernatant and resuspend the transformed cell pellet with the remaining supernatant (150 µL).17.Pipet 150 µL of resuspended cells per LB agar plate containing kanamycin (50 µg/mL).18.Spread cells on agar surface using sterile glass spreader.Critical step: Glass rods should be sterilized prior to spreading cells on LB-kan^+^ agar plates. Flame the glass rod for 3–5 s, dip into ethanol, flame again for 3–5 s, and then allow cooling for 10–12 s. Repeat this process between streaking samples.19.Incubate plates for 16–18 h at 37 °C.Critical step: To confirm that transformation was successful, pick a single colony from the transformation plate and inoculate in 5 mL LB-kan^+^ media broth. After overnight incubation at 37 °C with shaking at 220 rpm, use a miniprep kit to isolate DNA. DNA sequencing of this sample will determine whether the Quikchange PCR and transformation steps were successful (see Section 3.3).

### 3.3. PCR Screening and DNA Sequencing (Time of Completion: 1 Day)

20.Obtain transformation plate with colonies harboring mutant construct.21.Prepare 15 mL Falcon tubes (three for each mutant) with 5 mL LB-kan^+^ broth media in each.22.With an inoculating loop, and aseptic technique, pick a colony from the plate, and dip & swirl loop into a Falcon tube containing media for each. Do this step for three colonies.23.Incubate samples for 4–6 h at 37 °C with shaking at 220 rpm.Critical step: Following incubation, 1 µL incubated transformation product will be used in step 24, and the remained of the cultured 5 mL LB-kan^+^ broth media may be stored at 4 °C. This culture will be utilized following gel electrophoresis analysis to ensure proper DNA length (see step 29).24.Prepare PCR centrifuge tubes with 1 µL incubated transformation product, 1 µL forward primer (10 µM), 1 µL reverse primer (10 µM), 10 µL dNTP (10 mM), 1 µL Pfu Turbo DNA Polymerase (1 unit/50 µL), 5 µL 10× Pfu Turbo Cx Reaction Buffer, and dilute to 50 µL with deionized water (see Table 1).25.Run PCR thermo cycle to amplify transformed DNA sequence (see Figure 4).26.Prepare a 1.5% agarose gel.27.Load 30 µL of PCR product into 1.5% agarose gel, along with 1 kbp DNA ladder, and run at 60 V for 20 min.28.Analyze PCR product bands to confirm that the DNA sequence is both present at a reasonable concentration and the correct length.29.Spin down remainder of cultures for 10 min at 4 °C and 4000 rpm using Thermo Scientific Sorvall Legend RT+ Centrifuge.30.Discard supernatant and isolate DNA plasmid from pelleted cells using Thermo Scientific GeneJET Plasmid Miniprep Kit.31.Measure DNA miniprep product with Nanodrop. Expect concentration of 100–300 ng/µL.32.Prepare 500 ng of miniprep product in deionized water (10 µL total volume each).33.Each sample requires two tubes, one with forward primer (0.5 µL at 10 µM) and the other with reverse primer (0.5 µL at 10 µM) added.34.Send samples to DNA sequencing facility.35.Analyze results to ensure desired mutations are present before moving forward with protein expression.

### 3.4. Protein Expression (Time of Completion: 3 Days) 

#### 3.4.1. LB Media (Time of Completion: 2 h)

36.Add 25 g LB to 1 L of deionized water in 2 L Pyrex flask.Critical step: LB media should be used for KaiA and KaiB-FLAG expression. For FLAG-KaiC, M9 minimal media is highly recommended. To produce 1 L of M9 media, mix 6 g Na_2_HPO_4_, 3 g KH_2_PO_4_, 0.5 g NaCl, and 1 g NH_4_Cl, then autoclave on liquid cycle. Following sterilization, add 10 mL (0.2 µm filtered) 20% D-glucose, 2 mL 1 M MgSO_4_, 100 µL 1 M CaCl_2_, and 1 mL 50 µg/mL kanamycin.37.Autoclave on a liquid cycle to sterilize media (121 °C for 20 min).38.Allow to cool to 50 °C, and add 1 mL (50mg/mL) kanamycin to the sterilized 1 L LB media.

#### 3.4.2. Overexpressing Proteins in Cell Cultures (Time of Completion: 1 Day)

39.Pipet 5 mL LB kanamycin into sterilized 15 mL Falcon tubes for each variant protein that will be expressed.40.Add 2 µL of bacterial glycerol stock into corresponding falcon tubes.Critical step: All glycerol stocks consist of transformed cells in 16% glycerol. These solutions are prepared by adding 100 µL of 80% glycerol (autoclaved) to 400 µL of transformed cell media. These samples are stored at −80 °C until use.41.Incubate overnight in shaker for 16 h at 37 °C and 220 rpm.42.Add 5 mL overnight culture to autoclaved 1 L LB kanamycin media.43.Incubate at 37 °C and shake at 220 rpm until the optical density of 0.6 at 600 nm (OD_600_), measured with a UV-Vis Spectrometer, is reached.Critical step: After 4–6 h of incubation, optical density measurements of cell cultures at OD_600nm_ are carried out on a Hach DR 6000EDU UV-Vis spectrophotometer, with DI H_2_O used as a blank.44.Add IPTG to each flask for a final concentration of 200 µM.Critical step: IPTG induces protein expression in transformed *E. coli* by binding to *lac* operon repressor. This binding causes the repressor to dissociate from the DNA promoter region.45.Incubate at 25 °C with shaking at 220 rpm for 16 h.

#### 3.4.3. Harvesting Cells (Time of Completion: 2–3 h)

46.Distribute the 1 L cell culture into centrifuge bottles.47.Spin down at 5000 rpm for 10 min at 4 °C in a Thermo Scientific Sorvall RC 6 Plus Superspeed centrifuge.Critical step: For temperature controlled high speed/large volume centrifugation, a Thermo Scientific Sorvall RC 6 Plus Superspeed centrifuge with FiberLite F10-6x500y (500 mL bottles) and Sorvall Instruments SS-34 (50 mL bottles) rotors are used. All samples must be within 0.1 g of one another to ensure the rotors are balanced (steps 47 & 51).48.Discard supernatant and resuspend cell pellet with 30 mL lysis buffer (50mM NaH_2_PO_4_, 500mM NaCl, pH 8.0).Critical step: When preparing lysis buffer for cell pellet resuspension of FLAG-KaiC expressing cells, a final concentration of 1 mM ATP must be added to the buffer. FLAG-KaiC is a homohexameric protein with ATP molecules bound at protomer-protomer interfaces. Without ATP present KaiC is unstable and aggregates, reducing yield. ATP must also be present in the lysis buffer during cell cracking, in the nickel column wash and elution buffers, and in the mobile phase buffer during FPLC purification.49.Transfer resuspended samples to 50 mL Corning tubes. Here, the samples can be stored at −20 °C if needed for up to 48 h, but proceeding to cell lysis and protein purification (see 3.4.4) on the same day is preferred.

#### 3.4.4. Cell Lysis and Protein Purification (Time of Completion: 1 Day)

50.Lyse cells in homogenizer.51.Transfer the cell lysate into 50 mL centrifuge tubes and spin down at 15,000 rpm for 45 min.52.Prepare Ni-NTA columns for purification by running two full column volumes of deionized water through the nickel-chelated resin, followed by equilibration with a full column volume of lysis buffer.53.Load supernatant onto Ni-NTA columns.Critical step: Preceding the SUMO tag is a 6-His-tag that binds the Ni^2+^ ions, trapping the proteins of interest on the Ni-NTA column.54.Wash column with 50 mL wash buffer (50 mM NaH_2_PO_4_, 500 mM NaCl, 20 mM imidazole, pH 8.0) to remove non-His tagged proteins.55.Elute His tagged protein with 5 mL elution buffer (50 mM NaH_2_PO_4_, 500 mM NaCl, 250 mM imidazole, pH 8.0).Critical step: It is advised that 20 µL aliquots of supernatant, wash, and eluate be collected and analyzed by SDS-PAGE following Ni-NTA purification. This can help to quickly determine whether the protein of interest is in the eluate (desirable) or wash (undesirable).56.Add 150 µL of 100 µM ubiquitin-like-specific-protease 1 (Ulp1).Critical step: The Ulp1 enzyme specifically cleaves His_6_-SUMO tag from the fusion protein, allowing for purification by subsequent passes through Ni-NTA columns in which the His_6_-SUMO tag binds while the protein does not.57.Allow 8–14 h for Ulp1 to cleave His_6_-SUMO fusion proteins at 4 °C.58.Dilute the solution to 25 mM imidazole by adding lysis buffer to the sample.59.Clean Ni-NTA columns with two column volumes of DI water, followed by two bed volumes of 0.2 µm-filtered solution of 6 M guanidine hydrochloride (GdHCl) in 0.2 M acetic acid. Then wash with another two column volumes of DI water.60.Pass Ulp1-treated protein samples through clean Ni-NTA columns, and collect the flow-through in a clean, labeled corning tube.Critical step: When passing Ulp1-treated protein samples through Ni-NTA columns, it is important to regulate flow rates to ensure that the free His_6_-SUMO fusion tag can be captured on the Ni-NTA column, while proteins of interest pass through. This was performed by attaching a 1–2” section of soft rubber tubing to the bottom of each Ni-NTA column, which can then be partially clamped to allow flow rates of ~50–60 drop/min.61.Repeat steps 59 & 60 to ensure that protein sample has been sufficiently separated from His_6_-SUMO tags.Critical step: After cleavage of His_6_-SUMO by Ulp1 and purification by Ni-NTA columns, a confirmation step can be performed to verify that the protein of interest is present and pure. Transfer 3–5 µL of nickel column eluate to a centrifuge tube and add an equal volume of 2× SDS-dye. Load the sample alongside a protein ladder on SDS polyacrylamide gel (17% for KaiB) and run for 30 min at 60 V, followed by 100 min at 120 V. A comparison of stained bands against molecular weight markers allow for the estimation of protein purity and yield.62.Concentrate purified samples to 6 mL for subsequent FPLC injection (see 3.4.5) in an Amicon Stirred Cell concentrator using a 10 kDa molecular weight cut-off (MWCO) membrane.

#### 3.4.5. Fast Protein Liquid Chromatography (Time of Completion: 2–3 h)

63.Prepare mobile phase buffer in which the purified protein will be stored (20 mM Tris, 150 mM NaCl, pH 8, or pH 7.0 if the protein will be fluorescently labeled).Critical step: The preferred ratio of 6IAF to protein is 10–20:1, although we have had success at 5:1 fluorescently labeling KaiB. During the conjugation reaction the recommended pH is 7.0–7.5, which is accomplished by preparing the mobile phase for FPLC at this pH. All labeling was performed at 4 °C overnight in the dark.64.Divide the 6 mL of concentrated protein sample into 1 mL aliquots in six 1.5 mL centrifuge tubes and spin down at 13,000 rpm for 2 min on a benchtop centrifuge. This will allow for removal of any precipitate and particles from the sample before FPLC.65.Draw the supernatant from the centrifuge tubes into a 10 mL syringe using a long blunt-end needle.66.Exchange the needle on the syringe with an adaptor for connecting to the FPLC sample injection port.67.Remove all bubbles from the syringe.68.Attach the syringe to the sample injection port on the FPLC. Do so without introducing air into the instrument by filling the injection port with DI water as the syringe adaptor is inserted.69.Run the FPLC with HiLoad 16/600 Superdex 75 size-exclusion column, at 0.8 mL/min flow rate, and 0.5 MPa maximum pressure.70.Collect eluate peak fractions for proteins of interest.71.Following purification, concentrate eluate to desired concentration with an Amicon Stirred Cell concentrator using a 10 kDa molecular weight cut-off (MWCO) membrane.

### 3.5. Fluorophore Labeling of Clock Protein (Time of Completion: 1 Day)

72.Mix thiol-reactive fluorescent dye (1 mg 6IAF resuspended in 80 uL methanol) with protein at the ratio recommended for the particular dye being used.Critical step: 6IAF was chosen because it is inexpensive with high quantum yield, photostability, and moderate excitation/emission spectrum such that it may be paired as a FRET-acceptor or -donor in the future. As stated in step 63, a 5:1 ratio of 6IAF to protein is sufficient for ~99% labeling efficiency (by mass spectrometry, data not shown).73.Incubate at 4 °C for 14–16 h, covered to avoid light contamination.74.Use Amicon Stirred Cell concentrator with 10 kDa MWCO membrane to concentrate sample to 2 mL.75.Split the 2 mL sample into two 1.5 mL centrifuge tubes and spin down at 13,000 rpm for 2 min using a benchtop centrifuge.76.Repeat steps 65–68.77.Run the FPLC with HiPrep 26/10 Desalting Column, 3 mL/min flow rate, and 0.3 MPa max pressure to separate labeled protein from free fluorophore.78.Collect eluate from 11–19 mL.79.Following purification, concentrate eluate to desired molarity with the Amicon Stirred Cell concentrator and 10 kDa MWCO membrane.

### 3.6. Fluorescence Anisotropy Measurements (Time of Completion: 1–2 h)

80.Add reaction buffer (20 mM Tris, 150 mM NaCl, 5 mM MgCl_2_, 1 mM ATP, 0.5 mM EDTA, 50 µg/mL kanamycin, pH 8.0) to 1.5 mL centrifuge tubes.81.Add KaiA to a final concentration of 1.2 µM.82.Add unlabeled KaiB-FLAG to a final concentration of 3.45 µM.83.Adjust ATP concentration to 1 mM using ATP stock solution.84.Adjust MgCl_2_ concentration to 5 mM.85.Add FLAG-KaiC to a final concentration of 3.5 µM.86.Start 30-min timer, which indicates when to begin fluorimeter data collection.Critical step: During the 30-min timer duration, prepare the fluorimeter for data collection. Here, using 6IAF as a reporter, the monochromators on the PC1 excitation and emission wavelengths of 492 nm (slit width 2.0 nm) and 520 nm (slit width 1.0 nm), respectively, were read every 15 min. Fluorescence anisotropy measurements are achieved on the PC1 with an emission polarizer isolating parallel emission detection followed by subsequent perpendicular emission detection with the same photomultiplier tube (PMT). G-factor determination is set to the “once” option to adjust parallel and perpendicular gain values in order to eliminate instrumental bias in fluorescence anisotropy measurements.87.Load clock reaction samples into 1 mL syringe, attach 0.2 µm membrane filter, and filter samples into fresh 1.5 mL centrifuge tubes.Critical step: Filtering the solutions with low protein-binding 0.2 µm membrane ensures bacteria and/or large contaminants are not present in the final reaction mixtures. This step may be omitted if long term oscillations, many days/weeks, are not planned.88.Transfer into a fresh 1.5 mL centrifuge tube the volume of sample necessary to achieve a final volume of 600 µL upon adding fluorescently labeled protein in step 89.89.In the dark, add KaiB-FLAG-K25C-fluorophore to a final concentration of 0.05 µM and total volume of 600 µL.90.In the dark, mix the sample and transfer it to a fluorimeter cuvette.91.Place the cuvette in the fluorimeter and start data collection once the 30-min timer (see step 86) expires. Record the time representing t = 0.

## 4. Expected Results

Figure 5 shows real-time fluorescence anisotropy measurements of KaiB-FLAG-K25C-fluorophore over the course of 12 days. Note that robustness of the oscillation indicates that the experiment could have been extended considerably longer. Trough anisotropies are when labeled KaiB is mostly free in solution. In contrast, peak anisotropies are when KaiB is bound to KaiC and KaiA. The stability of the troughs is likely due to the stability of labeled KaiB. The slow decrease of peak anisotropies is likely due to a steady decrease in ATP and functional KaiC protein levels. In order to minimize potential artifacts due to the fluorophore, only 1.4% (50 nM) KaiB was labeled. Fluorescence intensities over 12 days decreased from 720,000 counts to 670,000, indicating that photobleaching is not a significant concern. A significant advantage of this protocol is that it allows hundreds of in vitro reactions to be monitored simultaneously in real time using a plate reader with minutes time resolution for weeks (this protocol has been employed to monitor oscillations on the Tecan Spark 10M and BMG CLARIOstar plate readers with nearly identical results relative to data collected on the PC1). Assuming that 15 ATP molecules are consumed per day per KaiC particle [23], after 12 days the ATP concentration is expected to be 0.3–0.4 mM. Note that the oscillator begins to fail around 0.1 mM ATP [23]. Although only KaiB was labeled in this protocol, KaiA and KaiC can in principle also be fluorescently labeled, although care must be taken regarding their naturally occurring cysteinyl residues. 

## Figures and Tables

**Figure 1 mps-02-00042-f001:**
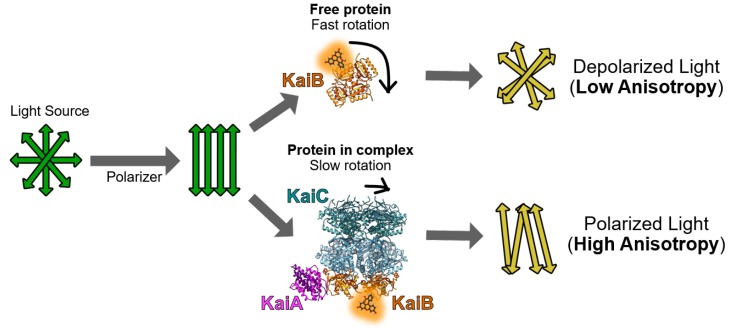
Cartoon of how labeled-KaiB protein free (**top**) and in ternary KaiA-KaiB-KaiC complexes (**bottom**) have significantly different fluorescence anisotropies. Rotational correlation times of the fluorophore can be extrapolated following a calibration of fluorescence anisotropy with neat fluorescein in the presence of glycerol, which has defined values [17].

**Figure 2 mps-02-00042-f002:**
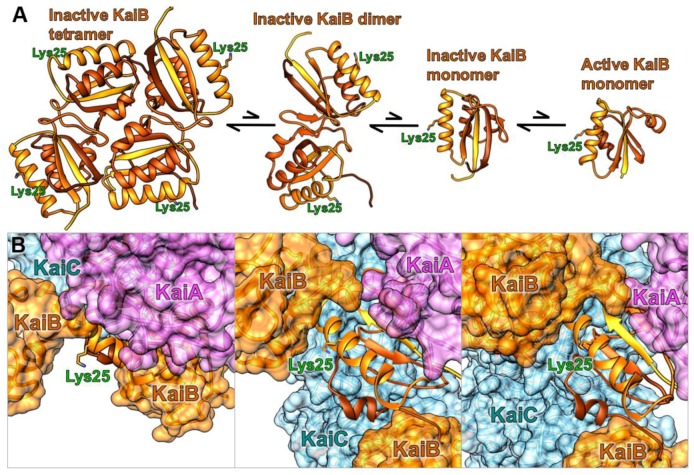
Fluorescence labeling site lysyl 25 on KaiB in relation to (**A**) KaiB’s four different free states and (**B**) ternary KaiABC complex shown in multiple orientations, with KaiA (purple), KaiB (orange), and KaiC (blue) [16].

**Figure 3 mps-02-00042-f003:**
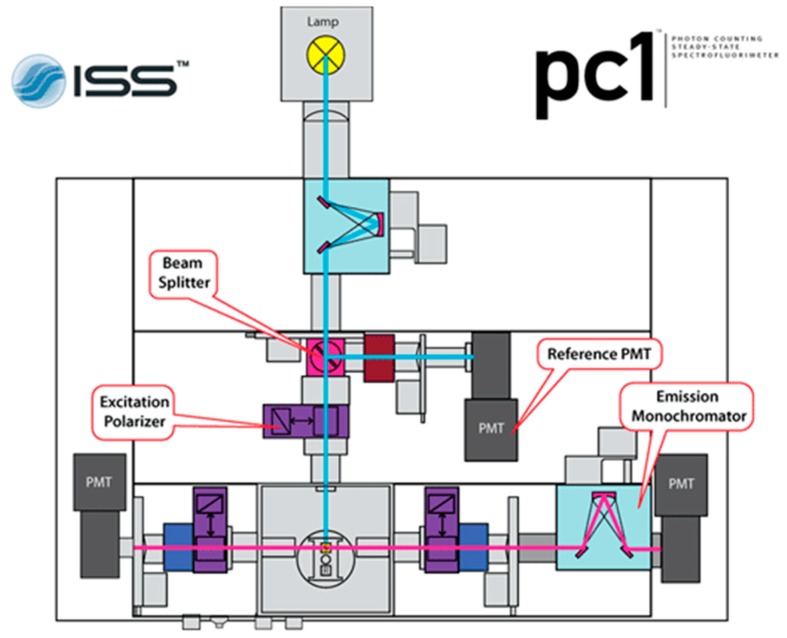
ISS PC1 instrument setup for polarization of incident light and detection of parallel and perpendicular polarized light after passing through the sample cuvette. This figure is from reference [21]. Here, once the sample is excited by polarized light, only the right instrument path is utilized, with the right photon multiplier tube collecting both parallel and perpendicular emissions in turn through software-controlled emission polarizer orientation adjustments (bottom right purple rectangle).

**Figure 4 mps-02-00042-f004:**
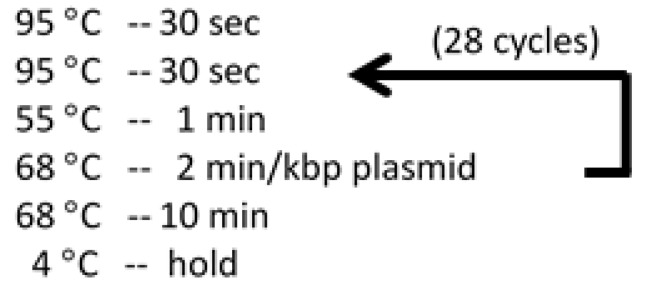
Temperatures and durations for Quikchange PCR thermo cycler program.

**Figure 5 mps-02-00042-f005:**
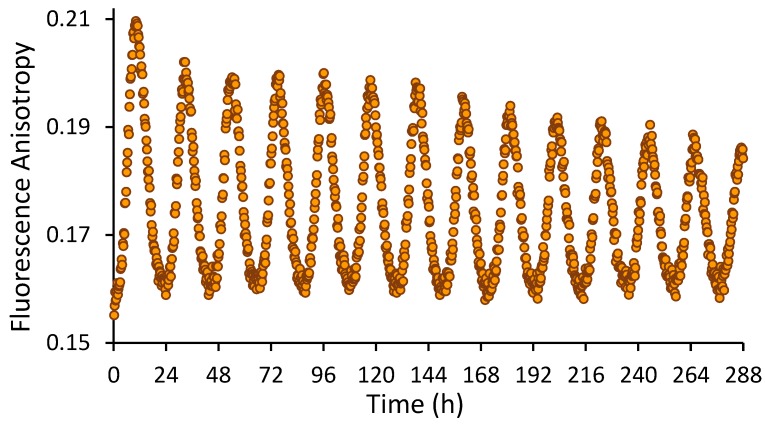
Fluorescence anisotropy of 0.05 µM KaiB-FLAG-K25C-6IAF in a reaction also containing unlabeled KaiA, KaiB-FLAG [20], and FLAG-KaiC at 1.2 µM, 3.45 µM, and 3.5 µM, respectively. Data points were collected every 15 min for 12 days.

**Table 1 mps-02-00042-t001:** Quikchange PCR reagents used in a generic reaction solution.

PCR Reaction	
5 µL (1×)	(10×) Pfu Turbo Cx Reaction Buffer
5 µL (50 ng)	Template Plasmid (10 ng/µL)
1 µL (200 nM)	Primer 1 (10 µM)
1 µL (200 nM)	Primer 2 (10 µM)
10 µL (2.5 mM)	dNTP Mix (10 mM)
27 µL	ddH_2_O
1 µL	Pfu Turbo DNA Polymerase (1 units/50 µL)
50 µL	Total Volume

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
