# Peer review of "Real-Time In Vitro Fluorescence Anisotropy of the Cyanobacterial Circadian Clock"

_mps, 2019, doi:10.3390/mps2020042_

Round 1
Reviewer 1 Report
Sufficient background & references:
1. the authors should explain that daily rhythms of complex formation among KaiA/B/C has already been reported and is the basis for implementing this very cool automated monitoring system. In this regard, these twp publications should be cited:
Kageyama H, Nishiwaki T, Nakajima M, Iwasaki H, Oyama T, Kondo T. Mol Cell. 2006 Jul 21;23(2):161-71. Cyanobacterial circadian pacemaker: Kai protein complex dynamics in the KaiC phosphorylation cycle in vitro.
Mori T, Williams DR, Byrne MO, Qin X, Egli M, Mchaourab HS, Stewart PL, Johnson CH. PLoS Biol. 2007 Apr;5(4):e93. Elucidating the ticking of an in vitro circadian clockwork.
2. The in vitro clock reaction monitored by fluorescence anisotropy (fluorescence polarization) has already been reported. This in no way lessens the importance of the current manuscript, but these two prior reports should be cited:
Leypunskiy E, Lin J, Yoo H, Lee U, Dinner AR, Rust MJ. Elife. 2017 Jul 7;6. pii: e23539. doi: 10.7554/eLife.23539. The cyanobacterial circadian clock follows midday in vivo and in vitro.
Chew J, Leypunskiy E, Lin J, Murugan A, Rust MJ. Nat Commun. 2018 Aug 1;9(1):3004. High protein copy number is required to suppress stochasticity in the cyanobacterial circadian clock.
3. 6-IAF–labeled KaiB-FLAG-K251C has been used in Welkie et al. (2018). The plasmid construct for expressing KaiB-FLAG-K251C described in the Sections 2.1.1 and 3.1 should be available to the scientific community. The authors should mention the availability of the material and reference: Welkie et al. 2018, Proc Natl Acad Sci USA 115: E7174-E7183.
4. The expression constructs for FLAG-KaiC, KaiA and KaiB-FLAG should be made available for research. Those constructs have been published by Cheng et al. (2015) and the others. The availability of the materials should be mentioned and reference: Chang et al. 2015 Science 349: 324-8.
Are the methods adequately described?
The authors should explain the rationale for using K25 as the labeling site. The authors should explain their rationale for using 6-IAF. (I.e., why 6-IAF rather than other dyes such as 5-IAF, 1,5-IAEDANS, etc.)
In Section 2.2.5 and Figure 3, the authors mentioned a specific instrument (ISS PC1). However, in Section 3.6 they don’t say anything about how to set up the instrument. What are the excitation and emission wavelengths? What are the slit widths? What are the gains for the PMT? What is the duration of FA measurements at each time point?
How is the software set up to do time course measurements?
How do the authors determine the Z-factor to calculate fluorescence anisotropy?
Is there any significant photobleaching of the 6-IAF signal during the 12-day time course?
In Section 3.5, a more detailed procedure should be included, e.g.:
What is the composition of the buffer solution?
What is the buffer? What is the pH?
Is TCEP in the labeling reaction? What is the concentration?
What concentration of KaiB-FLAG-K251C is in the labeling reaction?
What is the concentration of 6-IAF in the reaction?
How to prepare the stock/working solution of 6-IAF? What is a solvent (DMSO, DMF, or other)? What is the concentration of the 6-IAF stock solution? How should the 6-IAF solution be added to the reaction?
What is the labeling efficiency?
In Section 3.6, more detailed procedures should be included, e.g.:
In Step 87, why do the authors filter the reaction mixture through a 0.2-um membrane filter? Aren’t the protein concentrations changed by this filtration step?
How did the authors determine the concentration (50 nM) of 6-IAF–labeled KaiB-FLAG-K251C? Is this the best concentration to get good fluorescence signals, or is the optimal concentration to obtain excellent oscillations?
Are the results clearly presented?
The reactions contain 1 mM ATP and 3.5 uM KaiC. KaiC has ATPase activity at an average rate of ~15 ATP hydrolysis per KaiC per day. After 12 days, 0.63 mM ATP would be hydrolyzed and converted to ADP, and the ratio of ATP/(ATP+ADP) should go to 37%. However, the amplitude of the FA rhythms was reduced much less than expected by previous in vitro reaction experiments for a change of the ratio of ATP/(ATP+ADP) to 37%. Do the authors have an explanation?
Author Response
1. The authors should explain that daily rhythms of complex formation among KaiA/B/C has already been reported and is the basis for implementing this very cool automated monitoring system. In this regard, these two publications should be cited:
Kageyama H, Nishiwaki T, Nakajima M, Iwasaki H, Oyama T, Kondo T. Mol Cell. 2006 Jul 21;23(2):161-71. Cyanobacterial circadian pacemaker: Kai protein complex dynamics in the KaiC phosphorylation cycle in vitro.
Mori T, Williams DR, Byrne MO, Qin X, Egli M, Mchaourab HS, Stewart PL, Johnson CH. PLoS Biol. 2007 Apr;5(4):e93. Elucidating the ticking of an in vitro circadian clockwork.
The revision now cites the two papers and they are references #9 & #10.
2. The in vitro clock reaction monitored by fluorescence anisotropy (fluorescence polarization) has already been reported. This in no way lessens the importance of the current manuscript, but these two prior reports should be cited:
Leypunskiy E, Lin J, Yoo H, Lee U, Dinner AR, Rust MJ. Elife. 2017 Jul 7;6. pii: e23539. doi: 10.7554/eLife.23539. The cyanobacterial circadian clock follows midday in vivo and in vitro.
Chew J, Leypunskiy E, Lin J, Murugan A, Rust MJ. Nat Commun. 2018 Aug 1;9(1):3004. High protein copy number is required to suppress stochasticity in the cyanobacterial circadian clock.
The revision cites these two papers as references #12 & #13 as initial uses of this methodology. Please note that Leypunskiy et al. (2017) cites Heisler et al. (in press), which was in reference to this protocol that was shared.
3. 6-IAF–labeled KaiB-FLAG-K251C has been used in Welkie et al. (2018). The plasmid construct for expressing KaiB-FLAG-K251C described in the Sections 2.1.1 and 3.1 should be available to the scientific community. The authors should mention the availability of the material and reference: Welkie et al. 2018, Proc Natl Acad Sci USA 115: E7174-E7183.
The revision cites Welkie et al as reference #14, and states that this plasmid is available to the scientific community.
4. The expression constructs for FLAG-KaiC, KaiA and KaiB-FLAG should be made available for research. Those constructs have been published by Cheng et al. (2015) and the others. The availability of the materials should be mentioned and reference: Chang et al. 2015 Science 349: 324-8.
The revision now makes clear in section 2.1.5 that the plasmids described in Change (2015) are available upon request.
Are the methods adequately described?
The authors should explain the rationale for using K25 as the labeling site. The authors should explain their rationale for using 6-IAF. (I.e., why 6-IAF rather than other dyes such as 5-IAF, 1,5-IAEDANS, etc.)
The revision now states in section 1:
Figure 2 provides structural insight into the selected KaiB conjugation site for fluorophore labeling, highlighting how lysine 25 is oriented away from the intramolecular tetramer and dimer interfaces, and intermolecular KaiC and KaiA binding sites, allowing for conserved function of KaiB.
We tried multiple labeling sites along the same helix as K25. However, labeling at K25 was superior.
The revision now includes the following information in section 3.5:
6IAF was chosen because it is inexpensive with high quantum yield, photostability, and moderate excitation/emission spectrum such that it may be paired as a FRET-acceptor or -donor in the future. As stated in step 63, a 5:1 ratio of 6IAF to protein is sufficient for ~99% labeling efficiency (by mass spectrometry, data not shown).
In Section 2.2.5 and Figure 3, the authors mentioned a specific instrument (ISS PC1). However, in Section 3.6 they don’t say anything about how to set up the instrument. What are the excitation and emission wavelengths? What are the slit widths? What are the gains for the PMT? What is the duration of FA measurements at each time point?
How is the software set up to do time course measurements?
How do the authors determine the G-factor to calculate fluorescence anisotropy?
Is there any significant photobleaching of the 6-IAF signal during the 12-day time course?
The revision now includes the following information in section 3.6:
During the 30-minute timer duration, prepare the fluorimeter for data collection. Here, using 6IAF as a reporter, the monochromators on the PC1 excitation and emission wavelengths of 492 nm (slit width 2.0 nm) and 520 nm (slit width 1.0 nm), respectively, were read every 15 minutes. Fluorescence anisotropy measurements are achieved on the PC1 with an emission polarizer isolating parallel emission detection followed by subsequent perpendicular emission detection with the same photomultiplier tube (PMT). G-factor determination is set to the “once” option to adjust parallel and perpendicular gain values in order to eliminate instrumental bias in fluorescence anisotropy measurements.
The revision also now includes the following in section 4:
In order to minimize potential artifacts due to the fluorophore, only 1.4% (50 nM) KaiB was labeled. Fluorescence intensities over 12 days decreased from 720,000 counts to 670,000, indicating that photobleaching is not a significant concern.
In Section 3.5, a more detailed procedure should be included, e.g.:
What is the composition of the buffer solution?
What is the buffer? What is the pH?
Is TCEP in the labeling reaction? What is the concentration?
What concentration of KaiB-FLAG-K25C is in the labeling reaction?
What is the concentration of 6-IAF in the reaction?
How to prepare the stock/working solution of 6-IAF? What is a solvent (DMSO, DMF, or other)? What is the concentration of the 6-IAF stock solution? How should the 6-IAF solution be added to the reaction?
What is the labeling efficiency?
The revision now includes the following information in section 2.1.3:
Proteins can crosslink through cysteinyl –SH groups. To minimize this unwanted reaction, a final concentration 20 mM TCEP was added to the elution volume following Ni-NTA column purification. More TCEP can be added after the 2nd Ni-NTA column purification step, preceding concentration of the sample for FPLC (see steps 55-62). No additional TCEP is added following size-exclusion chromatography as to not interfere with 6IAF labeling efficiency.
The revision now includes the following information in section 2.1.4:
Here, for each fluorescently labeled protein preparation, 1 mg of 6IAF was suspended with 80 µL of methanol and then mixed in a 5:1 ratio with KaiB-FLAG-K25C at pH 7.0 to facilitate conjugation (see step 72).
After labeling, a desalting column is used to both adjust pH to 8.0 for oscillation reactions while also removing free dye from the sample.
• Desalting column buffer for protein storage (20 mM Tris, 150 mM NaCl, pH 8.0).
In Section 3.6, more detailed procedures should be included, e.g.:
In Step 87, why do the authors filter the reaction mixture through a 0.2-um membrane filter? Aren’t the protein concentrations changed by this filtration step?
Because we use low protein-binding filters, the concentrations of the proteins do not change significantly upon filtration.
The revision now includes the following information in section 3.6:
Filtering the solutions with low protein-binding 0.2 µm membrane ensures bacteria and/or large contaminants are not present in the final reaction mixtures. This step may be omitted if long term oscillations, many days/weeks, are not planned.
How did the authors determine the concentration (50 nM) of 6IAF–labeled KaiB-FLAG-K251C? Is this the best concentration to get good fluorescence signals, or is the optimal concentration to obtain excellent oscillations?
The concentration of labeled KaiB was determined empirically for optimum signal to noise while minimizing potential artifacts due to the fluorophore.
Are the results clearly presented?
The reactions contain 1 mM ATP and 3.5 uM KaiC. KaiC has ATPase activity at an average rate of ~15 ATP hydrolysis per KaiC per day. After 12 days, 0.63 mM ATP would be hydrolyzed and converted to ADP, and the ratio of ATP/(ATP+ADP) should go to 37%. However, the amplitude of the FA rhythms was reduced much less than expected by previous in vitro reaction experiments for a change of the ratio of ATP/(ATP+ADP) to 37%. Do the authors have an explanation?
Terauchi et al (reference #23) showed that even at 0.1 mM ATP KaiABC reactions oscillate for at least two days. Thus, it should not be too surprising that 0.37 mM ATP sustains a rather robust oscillation.
The revision also now includes the following in section 4:
Assuming that 15 ATP molecules are consumed per day per KaiC particle [23], after 12 days the ATP concentration is expected to be 0.3-0.4 mM. Note that the oscillator begins to fail around 0.1 mM ATP [23].
Reviewer 2 Report
In this manuscript the authors apply fluorescence anisotropy to study in vitro the protein interactions that determine the cyanobacterial clock. Thereto the clock proteins are fluorescently labeled. This is an interesting approach.
I have the following remarks concerning the fluorescence anisotropy determinations.
1. The authors do use the sensitive photon counting approach in a device from ISS. It is clear that the device must be properly used. This means that the polarization bias of the instrument at the selected wavelengths must be determined properly. I hope I did not overlook, but I was not able to find the wavelengths(bands) of excitation and emission that are used in the protocol. This must be specified, and this in combination with the steps for the determination of the polarization bias at these wavelengths.
2. Is the ISS instrument used with one or two detection channels? In Figure 3 only one emission monochromator is indicated. Is only one emission channel used?
3. What about the power of the light source used? Is the full power of the light source applied, or is an attenuation filter used?
4. In the Expected Results section, the authors refer to a plate reader approach. Did the authors evaluate this? The ISS instrument is very sensitive because of the photon counting method. Do plate readers offer the same sensitivity, and this in combination with anisotropy?
5. Can the authors interpret the higher and lower values of the fluorescence anisotropy in terms of the hydrodynamic volume of the rotating species? Do these values correspond with the expected size of the corresponding complexes?
6. Figure 5 shows a typical (and nice) result. I would appreciate a discussion by the authors concerning the stability of the lower values of the fluorescence anisotropy and the apparent systematic decrease of the higher values of the fluorescence anisotropy.
Author Response
In this manuscript the authors apply fluorescence anisotropy to study in vitro the protein interactions that determine the cyanobacterial clock. Thereto the clock proteins are fluorescently labeled. This is an interesting approach.
I have the following remarks concerning the fluorescence anisotropy determinations.
1. The authors do use the sensitive photon counting approach in a device from ISS. It is clear that the device must be properly used. This means that the polarization bias of the instrument at the selected wavelengths must be determined properly. I hope I did not overlook, but I was not able to find the wavelengths(bands) of excitation and emission that are used in the protocol. This must be specified, and this in combination with the steps for the determination of the polarization bias at these wavelengths.
The revision now includes the following information in section 3.6:
During the 30-minute timer duration, prepare the fluorimeter for data collection. Here, using 6IAF as a reporter, the monochromators on the PC1 excitation and emission wavelengths of 492 nm (slit width 2.0 nm) and 520 nm (slit width 1.0 nm), respectively, were read every 15 minutes. Fluorescence anisotropy measurements are achieved on the PC1 with an emission polarizer isolating parallel emission detection followed by subsequent perpendicular emission detection with the same photomultiplier tube (PMT). G-factor determination is set to the “once” option to adjust parallel and perpendicular gain values in order to eliminate instrumental bias in fluorescence anisotropy measurements.
2. Is the ISS instrument used with one or two detection channels? In Figure 3 only one emission monochromator is indicated. Is only one emission channel used?
The revision now includes the following information in section 2.2.5, Figure 3 caption:
ISS PC1 instrument setup for polarization of incident light and detection of parallel and perpendicular polarized light after passing through the sample cuvette. This figure is from reference [21]. Here, once the sample is excited by polarized light, only the right instrument path is utilized, with the right photon multiplier tube collecting both parallel and perpendicular emissions in turn through software-controlled emission polarizer orientation adjustments (bottom right purple rectangle).
3. What about the power of the light source used? Is the full power of the light source applied, or is an attenuation filter used?
The revision now includes the following information in section 2.2.5:
ISS Photon Counting Spectrofluorimeter (PC1), with 300 W xenon arc lamp attenuated such that fluorescently-labeled samples yield 600,000 - 800,000 photon counts, was used for all fluorescence measurements described here. This steady-state fluorimeter was setup with a three-stage cuvette system and water bath to control sample temperature. Excitation and emission wavelengths are easily manipulated, allowing for measurements of multiple fluorescent dyes simultaneously.
4. In the Expected Results section, the authors refer to a plate reader approach. Did the authors evaluate this? The ISS instrument is very sensitive because of the photon counting method. Do plate readers offer the same sensitivity, and this in combination with anisotropy?
In our hands the ISS instrument, Tecan Spark 10M, and BMG CLARIOstar all provide outstanding and consistently similar sensitivities using the protocol described here.
The revision now includes the following information in section 4:
A significant advantage of this protocol is that it allows hundreds of in vitro reactions to be monitored simultaneously in real time using a plate reader with minutes time resolution for weeks (this protocol has been employed to monitor oscillations on the Tecan Spark 10M and BMG CLARIOstar plate readers with nearly identical results relative to data collected on the PC1).
5. Can the authors interpret the higher and lower values of the fluorescence anisotropy in terms of the hydrodynamic volume of the rotating species? Do these values correspond with the expected size of the corresponding complexes?
In principle, fluorescence anisotropy can be used to estimate the hydrodynamic volume of the protein. However, this is complicated given the possible additional rotational degrees of freedom of the fluorophore. Regardless, the rotational correlation time of the fluorophore can be estimated.
The revision now includes the following information in section 1, Figure 1 caption:
Rotational correlation times of the fluorophore can be extrapolated following a calibration of fluorescence anisotropy with neat fluorescein in the presence of glycerol, which has defined values [17].
6. Figure 5 shows a typical (and nice) result. I would appreciate a discussion by the authors concerning the stability of the lower values of the fluorescence anisotropy and the apparent systematic decrease of the higher values of the fluorescence anisotropy.
The revision now includes the following information in section 4:
Trough anisotropies are when labeled KaiB is mostly free in solution. In contrast, peak anisotropies are when KaiB is bound to KaiC and KaiA. The stability of the troughs is likely due to the stability of labeled KaiB. The slow decrease of peak anisotropies is likely due to a steady decrease in ATP and functional KaiC protein levels.